# Pharmacists’ Perceptions and Drivers of Immunization Practices for COVID-19 Vaccines: Results of a Nationwide Survey Prior to COVID-19 Vaccine Emergency Use Authorization

**DOI:** 10.3390/pharmacy9030131

**Published:** 2021-07-26

**Authors:** Sonia Susan Jacob, Mary Barna Bridgeman, Hyoeun Kim, Michael Toscani, Racquel Kohler, Stephanie Shiau, Humberto R. Jimenez, Joseph A. Barone, Navaneeth Narayanan

**Affiliations:** 1Rutgers Pharmaceutical Fellowship Program, Rutgers University, Piscataway, NJ 08854, USA; miket@pharmacy.rutgers.edu; 2Department of Pharmacy Practice and Administration, Ernest Mario School of Pharmacy, Rutgers University, New Brunswick, NJ 08901, USA; mary.bridgeman@pharmacy.rutgers.edu (M.B.B.); hk607@scarletmail.rutgers.edu (H.K.); umbe@pharmacy.rutgers.edu (H.R.J.); jbarone@pharmacy.rutgers.edu (J.A.B.); navan12@pharmacy.rutgers.edu (N.N.); 3Cancer Institute of New Jersey, Rutgers, The State University of New Jersey, New Brunswick, NJ 08901, USA; kelly.kohler@rutgers.edu; 4Department of Biostatistics and Epidemiology, School of Public Health, Rutgers University, Piscataway, NJ 08854, USA; stephanie.shiau@rutgers.edu

**Keywords:** COVID-19, vaccines, vaccine acceptance, vaccine hesitancy, pharmacists, perceptions

## Abstract

Background: Pharmacists play a vital role in recommending and providing vaccines to improve public health and are on the front line of mass immunization efforts. Aim: The objective of this study is to evaluate pharmacists’ perceptions on COVID-19 vaccines prior to emergency use authorization (EUA) amid a global pandemic. Methods: A voluntary, anonymous, cross-sectional survey was conducted between September and November 2020. Survey respondents included a convenience sample of licensed pharmacists in the United States. The primary outcomes were pharmacists’ willingness to receive and recommend hypothetical COVID-19 vaccines. Covariates assessed in the survey included COVID-19 exposure or personal experience, primary pharmacy practice setting, background in training, geographic region, and prioritization of clinical data. The data were analyzed using descriptive and inferential statistics. Results: This study surveyed 763 pharmacists and results from 632 participants were included in final analysis. Overall, 67.1% of the pharmacists were willing to receive a COVID-19 vaccine and 63.4% of the pharmacists were willing to recommend a COVID-19 vaccine at ≤1 year from the time of vaccine approval. At >1 year after vaccine approval, 78% of the pharmacists were willing to receive a COVID-19 vaccine and 81.2% of the pharmacists were willing to recommend a COVID-19 vaccine. Conclusions: Survey findings suggest that, while a majority of pharmacists surveyed indicate acceptance of hypothetical COVID-19 vaccines, there remains to be hesitancy among pharmacists to receive or recommend vaccination.

## 1. Introduction

Since January of 2020, the novel coronavirus (COVID-19) pandemic has swept across the United States, resulting in more than 522,973 deaths as of 8 March 2021 [1]. Global research initiatives to determine optimal clinical management strategies are ongoing, but much remains unknown. To improve public health, the White House executed Operation Warp Speed in March 2020, which promoted the development and manufacturing of COVID-19 vaccines by selected pharmaceuticals in order to accelerate the development of novel therapeutic agents to aid the response to this pandemic throughout the country [2]. Despite the efforts to release a COVID-19 vaccine in a timely manner, a Reuters poll conducted in May 2020 demonstrated that one-fourth of Americans were uninterested in getting the coronavirus vaccine once available. The common reasons included nervousness about the vaccine’s quick approval and weariness about the risk associated with a new vaccine [3]. It is important to note that the general public’s knowledge regarding COVID-19 may have been influenced by myths circulating on social media [4]. However, another survey revealed that participants were more willing to get vaccinated if they thought their healthcare provider would recommend vaccination [5]. Pharmacists are on the frontline of mass immunization efforts and play a vital role in recommending vaccines to improve public health. This involvement also makes pharmacists at increased risk for exposure, thus, emphasizing the importance to receive a COVID-19 vaccine. The Department of Health and Human Services authorized pharmacists to order and administer COVID-19 vaccines under the Public Readiness and Emergency Preparedness (PREP) Act [6]. Previous research has shown that healthcare providers with high confidence in the benefits and safety of vaccines were more likely to accept vaccines for themselves and recommend vaccines to their patients [7].

In April 2020, there were 78 active COVID-19 vaccine candidates. These vaccine candidates include the Pfizer Inc. and Moderna Inc. mRNA vaccine platforms and the Janssen viral vector vaccine platform [8]. The preliminary updates on the efficacy of COVID-19 vaccines in phase III clinical trials may have influenced many pharmacists’ perceptions within a short period of time. In a survey conducted among 400 pharmacists in November 2020, 52% of pharmacists were ready to receive the COVID-19 vaccine as soon as possible, 74% were willing to administer the vaccine as soon as possible, and 37% were willing to receive and vaccinate patients as soon as possible [9]. In December 2020, the FDA’s Vaccines and Related Biological Products Advisory Committee (VRBPAC) approved the United States’ first COVID-19 vaccines Emergency Use Authorization (EUA) submitted by Pfizer Inc. and Moderna Inc. [10,11]. On 27 February 2021 the FDA’s VRBPAC approved the Janssen COVID-19 vaccine under emergency use authorization [12]. The COVID-19 vaccine global landscape continues to expand—with 108 vaccines in clinical development and 3 vaccines approved for emergency-use authorization by the Food and Drug Administration (FDA) in the United States as of 15 July 2021 [13].

It is crucial for pharmacists to stay informed on COVID-19 vaccines’ safety and efficacy data to address vaccine hesitancy so that their patients can feel comfortable in deciding to receive the vaccine. Through this study, we aim to assess how clinical information, background training, and personal experience with COVID-19, as well as how pharmacists’ knowledge, beliefs, and attitudes influence their perceptions toward COVID-19 vaccines. The objective of this study was to assess what factors influence pharmacists’ willingness to receive and recommend COVID-19 vaccines and to identify pertinent information that pharmacists need to make an informed decision to receive and recommend COVID-19 vaccines.

## 2. Materials and Methods

This was a national cross-sectional study administered with use of the secure online survey platform, Qualtrics. A convenience sample of voluntary participants were recruited nationally through email- and listserv-based distribution, utilizing publicly accessible contact lists in pharmacy professional organization listservs and professional social media platforms such as LinkedIn. The professional organization listservs that were used include Rutgers pharmacy preceptors, Rutgers Pharmaceutical Industry Fellowship (RPIF) current fellows and alumni, Garden State Pharmacy Owners, New Jersey Society of Health-System Pharmacists (NJSHP), American College of Clinical Pharmacy (ACCP), American Society of Health-System Pharmacists (ASHP), American Society of Consultant Pharmacists (ASCP), and American Association of Colleges of Pharmacy (AACP). Once participants agreed with the informed consent form, they were able to access the 34-question survey. The survey was open on 18 September 2020 and closed on 16 November 2020. This survey data was collected prior to any preliminary COVID-19 vaccine data.

Any licensed U.S. pharmacist from various settings was able to participate. The exclusion criteria included pharmacy technicians, interns, externs, and non-pharmacy licensed subjects. Participants were excluded from analysis if they did not consent to the survey, were not a practicing pharmacist in the USA, completed ≤ 15% of the survey, or did not respond to the questions corresponding to the primary outcome. The study variables collected were parameters in the form of yes/no questions, select all that apply, and ordinal scale ranking. The independent variables collected were demographics, exposure to COVID-19, work type, likelihood to receive a vaccine, likelihood to recommend a vaccine, concerns of vaccine safety, concerns of vaccine efficacy, and prioritization of vaccine information.

The primary outcomes were the pharmacists’ willingness to receive and recommend COVID-19 vaccines. This was assessed using Questions 22, 23, 26, and 27 (Appendix A and the use of a Likert scale). We created a binary variable indicating willingness to receive and recommend a COVID-19 vaccine for “somewhat likely” or ‘extremely likely” responses. Pharmacists’ willingness to receive and recommend COVID-19 vaccines were assessed at ≤ 1 year from the time of vaccine approval and > 1 year from the time of vaccine approval. Covariates assessed in the survey included COVID-19 exposure or personal experience, primary pharmacy practice setting, background in training, region of the United States, and prioritization of clinical data. Pharmacists’ perceptions of COVID-19 and COVID-19 vaccines were also assessed. Further exploratory analyses were done to assess concordance and discordance among pharmacists’ decisions to receive and recommend using Questions 22, 23, 26, and 27 (Appendix A).

This survey was approved by the Institutional Review Board of Rutgers University and data were analyzed using descriptive and inferential statistics. The *p*-values were determined using a chi-squared analysis. All analyses were conducted using SAS 9.4 (Cary, NC, USA).

## 3. Results

A total of 763 participants responded to the survey, however, only 632 participants were included in the final analysis. A total of 131 participants were excluded because 6 (4.6%) did not consent, 61 (46.6%) were not practicing pharmacists in the USA, 41 (31.3%) completed ≤ 15% of the survey, and 23 (17.6%) of the participants did not answer one or more of the four prime study questions regarding their willingness to receive or recommend a COVID-19 vaccine.

Characteristics of the study population are shown in Table 1. The majority of our respondents were from the Northeast region of the USA, female, between 18 and 55 years old, white, worked in a direct patient care setting, and had completed post-doctoral training.

Perceptions of COVID-19 and new COVID-19 vaccines were assessed and are presented in Table 2. The majority of pharmacists were concerned about becoming infected with COVID-19, believed it would affect their daily work, and believed spread was dependent on human control. However, only about half of the pharmacists believed that a COVID-19 vaccine would protect them.

Pharmacists’ willingness to receive and recommend a COVID-19 vaccine were assessed at ≤ 1 year from the time of vaccine approval and > 1 year after vaccine approval, as listed in Table 2. Pharmacists were significantly more likely to receive and recommend a COVID-19 vaccine at > 1 year after vaccine approval as compared were ≤ 1 year after vaccine approval. Overall, more than half of the pharmacists were somewhat likely or extremely likely to receive and recommend a COVID-19 vaccine.

Perceptions of COVID-19 and new COVID-19 vaccines were assessed and are presented in Table 3. The majority of pharmacists were concerned about becoming infected with COVID-19, believed it would affect their daily work, and believed spread was dependent on human control. However, only about half of the pharmacists believed that that a COVID-19 vaccine would protect them.

A secondary analysis was conducted to determine how consistent individual respondents were in their choice to receive and recommend COVID-19 vaccines When comparing the responses for vaccine decision-making to receive vs. recommend a COVID-19 vaccine at ≤ 1 year after vaccine approval, significantly more pharmacists were “somewhat likely or extremely likely,” to recommend the vaccine than to receive the vaccine (*p* < 0.0001). Pharmacists’ concordance between vaccine receival and recommendation status showed that 59.7% of the pharmacists said they were somewhat likely/extremely likely to both receive and recommend COVID-19 vaccines and 29.1% of the pharmacists said they were neither likely nor unlikely/somewhat unlikely/or extremely unlikely to both receive and recommend COVID-19 vaccines. When assessing discordance, it was found that 3.8% of the pharmacists that said they were “somewhat likely/extremely likely to recommend the vaccine” but answered that they were “neither likely nor unlikely, somewhat unlikely, or strongly unlikely,” to receive the vaccine themselves at ≤ 1 year from time of vaccine approval.

A comparison of responses for vaccine decision-making to receive a COVID-19 vaccine vs. to recommend a COVID-19 vaccine was also completed at > 1 year after vaccine approval. There was a statistically significant difference when pharmacists were asked if they would receive a COVID-19 vaccine vs. recommend a COVID-19 vaccine at > 1 year after vaccine approval (*p* < 0.0001). Pharmacists’ concordance between vaccine receival and recommendation status >1 year after vaccine approval showed that 72.8% of the pharmacists said they were “somewhat likely/extremely likely to both receive and recommend COVID-19 vaccines” and 13.6% of the pharmacists said they were “neither likely nor unlikely/somewhat unlikely/or extremely unlikely to both receive and recommend COVID-19 vaccines”. When assessing discordance, it was found that 8.4% of the pharmacists that said they would “somewhat agree or strongly agree,” to recommend the vaccine, but answered that they were “neither likely nor likely, somewhat disagree, or strongly disagree,” to receive the vaccine themselves >1 year after vaccine approval.

Potential factors that may influence pharmacists’ decision to receive a COVID-19 vaccine are illustrated in Table 4. There was a significant difference (*p* < 0.0001) between collective responses of each of the five categories (extremely unlikely, somewhat unlikely, neither likely nor unlikely, somewhat likely, and extremely likely) at ≤ 1 year and > 1 year after vaccine approval.

Potential factors that may influence pharmacists’ decision to recommend a COVID-19 vaccine are illustrated in Table 5. There was a significant difference (*p* < 0.0001) between collective responses of each of the five categories (extremely unlikely, somewhat unlikely, neither likely nor unlikely, somewhat likely, extremely likely) at ≤ 1 year and > 1 year after vaccine approval.

If participants chose “neither unlikely nor likely, somewhat unlikely, or extremely unlikely,” to either receive or recommend COVID-19 vaccines they were then asked to select all reasons for why they would delay or refuse a COVID-19 vaccine, regardless of time of vaccine approval. Participants who said they would delay or refuse receiving a COVID-19 vaccine (*n* = 253) chose to do so for the following reasons: concerns about side effects or sickness (81%), rapidly changing real world data during pandemic (46.6%), not in a high-risk or priority group (23.7%), belief that the vaccine does not work (23.3%), limited contact with high-risk patients (12.6%), ability to social distance and wear a mask provides sufficient protection (5.5%), COVID-19 is non-severe or temporary issue (3.2%), vaccine candidate already had COVID-19 (0.8%), COVID-19 vaccination is not needed nor wanted (6.3%), and other (28.9%). If participants chose “somewhat agree or strongly agree,” to any of Question 23, 24, 26, or 27, then, they were asked to select all reasons they chose to receive or recommend the vaccine. Participants who said they would delay or refuse recommending a COVID-19 vaccine (*n* = 253) to their patients, chose to do so for the following reasons: concerns about side effects or sickness (75.1%), rapidly changing real world data during pandemic (41.1%), belief that the vaccine does not work (17%), not in a high-risk or priority group (15%), limited contact with high-risk patients (9.5%), hesitant about vaccine discussion with patients (9.1%), ability to social distance and wear a mask provides sufficient protection (4.7%), vaccine candidate already had COVID-19 (4.7%), COVID-19 is a non-severe or temporary issue (2%), other (20.6%). Concerns that fell under the “other,” category included political influence on vaccine approval, and uncertainty regarding the safety and efficacy data, and fear that the approval process would be too quick or compromise safety.

If participants chose “somewhat likely or extremely likely,” to either receive or recommend COVID-19 vaccines. Then, they were asked to select all reasons why they were in favor of the COVID-19 vaccine. Among those who favored the COVID-19 vaccine for themselves (*n* = 565), the following reasons included: avoid transmitting COVID-19 to family (91.9%), reduce the risk of contracting COVID-19 to protect self (87.6%), being a healthcare worker (80%), avoid transmitting COVID-19 to community or patients (64.1%), workplace mandate (47.6%), recommendations of government health agencies (43.5%), recommendation from other medical professionals (40.4%), have a medical condition that puts vaccine candidate at increased risk for COVID-19 (12.9%), recommendations of friends and family (9%), recommendations of political leaders (1.8%), other (41.6%). Among those who favored recommending the COVID-19 vaccine to their patients (*n* = 537), the following reasons included: reduce the risk of contracting COVID-19 to protect self (94.2%), avoid transmitting COVID-19 to family (90.5%), avoid transmitting COVID-19 to community or patients (90.3%), patient has a medical condition that puts them at increased risk for COVID-19 (86%), recommendations of government health agencies (62.9%), recommendations of political leaders (2.2%), and other (1.7%).

Regardless of the pharmacists’ decision to receive or recommend vaccination, all respondents were asked to prioritize the top three clinical pieces of clinical information they would prioritize when formulating a decision to receive or recommend a COVID-19 vaccine. The prioritization of this clinical information is shown in Figure 1.

## 4. Discussion

To date, this is the largest survey study to report the results of pharmacists’ perceptions of and willingness to receive and recommend COVID-19 vaccines during a global pandemic. A previous study conducted by Petek et al. showed that familiarity with influenza disease did not impact the decision to vaccinate for influenza [14]. Pharmacists’ perceptions of COVID-19 were assessed to gauge if the rapidly changing knowledge of the disease state would influence pharmacists’ view of credibility for a new vaccine. Our study found that 67.1% of the pharmacists were willing to receive a COVID-19 vaccine and 63.4% of the pharmacists were willing to recommend a COVID-19 vaccine at ≤ 1 year from the time of vaccine approval. At > 1 year after vaccine approval, 78% of the pharmacists were willing to receive a COVID-19 vaccine and 81.2% of the pharmacists were willing to recommend a COVID-19 vaccine. There was a significant difference in pharmacists’ vaccine decisions between the varying time frame of vaccine approval. Pharmacists with post-doctoral training were more likely to receive and recommend a COVID-19 vaccine than those without post-doctoral training. Pharmacists in indirect patient care settings were more likely to receive or recommend a COVID-19 vaccine than those in direct patient care settings. Lastly, pharmacists with increased personal experience and exposure to COVID-19 were more likely than unlikely to receive and recommend a COVID-19 vaccine. Pharmacist with no post-doctoral training were less likely to receive or recommend a COVID-19 vaccine than those with post-doctoral training. Pharmacist in direct patient care settings were less likely to receive or recommend a COVID-19 vaccine than those indirect patient care settings. Pharmacist without personal experience nor exposure to COVID-19 were more unlikely than likely to receive or recommend a COVID-19 vaccine. These determinants may aid in identifying current concerns of pharmacists so that medical education may be tailored to address those concerns.

Our results were consistent with the results of another nationwide survey of 400 pharmacists conducted by the American Pharmacist Association (APhA) from 21 November 2020 to 28 November 2020. The survey conducted by APhA showed that 69% of pharmacists would receive a COVID-19 vaccine at ≤ 1 year and 60% of pharmacists would recommend the vaccine at ≤ 1 year [9]. The APhA survey was conducted shortly after preliminary results for COVID-19 vaccines efficacy trials were published [15,16]. Our survey was conducted from 19 September 2020 to 18 November 2020 (prior to release of any COVID-19 vaccine trial data) and showed that baseline willingness status of pharmacists to receive and recommend COVID-19 vaccines were similar to their willingness status once the preliminary clinical trial findings were released. A major difference between our study and the survey conducted by APhA is that the APhA survey assessed pharmacists’ willingness at three different time intervals: vaccination as soon as possible, after six months of experience with the vaccine, and after one year of experience with the vaccine. Although we assessed willingness at ≤ 1 year and > 1 year of vaccine approval, this difference between studies may be negligible in a real-world setting. COVID-19 vaccine distribution is decided by phases recommend by the Centers of Disease Control and Prevention; therefore, recipients may not have the ability to choose when they will receive the vaccine within the first year of vaccine roll-out. The APhA survey also assessed how prepared pharmacist felt to administer COVID-19 vaccines as soon as they are available. Pharmacists had the option of stating the following: they are ready to receive and vaccinate as soon as the vaccine is available, they are in the final preparation stages and are ready to vaccinate patients as soon as the vaccine is available, they are not prepared to vaccinate in the first wave of the vaccination effort but plan to participate in future phases, or logistics of preparation are too difficult for them to participate [9]. This is a component that our survey did not assess but would be useful to determine what resources pharmacists need to feel prepared, in order to increase their willingness to receive and recommend vaccines.

Previous research has found that 69% of the general U.S. population were willing to get a COVID-19 vaccine. Reiter et al. conducted a cross-sectional survey which consisted of 2006 adults ages 18 and older during May 2020. Reiter et al. demonstrated that participants were more likely to get vaccinated if they thought their healthcare provider would recommend vaccination. The general U.S. population was also more likely to be willing to get vaccinated if they reported higher levels of perceived risk of being infected by COVID-19, or perceived effectiveness of COVID-19 vaccine. The results from the general adult U.S. population also revealed that participants were less likely to be willing to get vaccinated if they were non-Latinx Black or had higher levels of perceived potential vaccine harm [5]. The rate of willingness to receive a COVID-19 vaccine was comparable to other studies done surveying the general adult population [5]. Since the general adult population consider healthcare providers as a major influencer in their vaccine decision, the need to increase pharmacists’ confidence and acceptance of COVID-19 vaccines is evident. We also did not analyze how willingness toward vaccination differed between race, although this may be important for future findings to address perceptions as they relate to health disparities.

An online survey of U.S. adults observed trends in vaccine trust and vaccine hesitancy between 14 October 2020 and 29 March 2021. Daly et al. surveyed 7420 participants who provided 42,154 survey responses. U.S adults showed that overall vaccine hesitancy has declined significantly by 10.8% from October 2020 (46%) to March 2021 (35.2%). U.S. adults were asked to rate their trust for the “governmental approval process to ensure the COVID-19 vaccine is safe for the public,” and the “process in general (not just for COVID-19) to develop safe vaccines for the public,” on a scale of one (fully trust) to four (do not trust). Change in trust increased by 0.4 from October 2020 (3.0) to March 2021 (2.6) [17]. Since our survey was conducted before vaccine data showed high efficacy against COVID-19, there is a need to conduct a survey of pharmacists’ trust and willingness to vaccinate against COVID-19 in the USA, now that COVID-19 vaccines have regulatory approval and the existence of mass immunization programs.

Research is still needed to understand how U.S. pharmacists’ readiness to provide COVID-19 services impacts their willingness to provide COVID-19 vaccination. Merks et al. conducted a cross-sectional survey that consisted of 1777 Polish community pharmacist between February and August 2020. Merks et al. assessed the readiness and willingness of pharmacists, trained and not trained through a pilot immunization workshop, to provide vaccination services during the COVID-19 pandemic in Poland. Polish community pharmacists who underwent additional immunization training showed a greater readiness to provide immunization services as compared with those who did not receive training. The most common barriers affecting Polish pharmacists’ readiness were: providing vaccinations will require more work, not enough training courses, and pharmacy facilities are not adjusted to provide these services [18]. Acknowledging that readiness and willingness for vaccination are not mutually exclusive, our findings were comparable in showing that those who have completed additional training were more likely than unlikely to receive and recommend COVID-19 vaccines. This may emphasize the effects of educational training on vaccine acceptance. Our study did not analyze what resources are needed to overcome vaccine hesitancy in pharmacists or barriers in vaccine implementation in pharmacy practice settings.

There are several limitations to our study. First, the use of a convenience sample may have allowed for possible selection bias in regard to survey distribution. According to the Bureau Labor of Statistics, 42% of the pharmacy graduates work in a community practice setting and 26% of the pharmacy graduates work in a hospital practice setting, therefore, our study population is not representative of the pharmacist workforce data in the United States [19]. Moreover, there was no standardized denominator to accurately determine the survey response rates given the broad distribution across online platforms and email distribution lists from professional organizations. Lastly, our survey took place before multiple companies filed for the FDA Emergency Use Authorization which publicized the ~95% effectiveness clinical trial data [14,15]. This vaccine data could certainly affect the perceptions of pharmacists toward COVID-19 vaccines and the current survey findings may change based on the now available data.

However, a potential strength of our data is that it was taken before the interim COVID-19 vaccine results and can be viewed as pharmacists’ unbiased baseline perceptions. Our data represents how pharmacists feel at baseline and how pharmacists are influenced prior to release of vaccines emergency use authorization. This not only informs the current pandemic, but future ones as well. Lastly, to the best of our knowledge, this is the largest survey assessing pharmacist personal and professional behaviors regarding future COVID-19 vaccines. There were survey respondents across the country as well as across specialty areas and practice settings.

Vaccines are among the most effective public health interventions worldwide. By understanding possible determinants in vaccine decision-making, we should be able to further tailor vaccine education and understand barriers to vaccine acceptance. Further research is needed on both health behaviors and vaccine confidence among healthcare professionals, as well as clinical decision-making in areas of novel therapeutics.

## 5. Conclusions

This cross-sectional survey was conducted from September to November 2020 (prior to the release of any COVID-19 vaccine trial data) but during the time governing bodies such as the Department of Health and Human Services, Food and Drug Administration, and the White House issued COVID-19 vaccine emergency use authorization guidance. Pharmacists, as members of the front-line immunization effort, play a vital role in recommending and providing vaccines to improve public health. We aimed to assess pharmacists’ willingness to receive and recommend COVID-19 vaccines and factors that may influence this decision. Our study showed that although most pharmacists would receive and recommend a COVID-19 vaccine, a substantial proportion of pharmacists reports vaccine hesitancy to receive or recommend COVID-19 vaccines. Pharmacists prioritize similar clinical information when considering the decision to receive and recommend a COVID-19 vaccine.

## Figures and Tables

**Figure 1 pharmacy-09-00131-f001:**
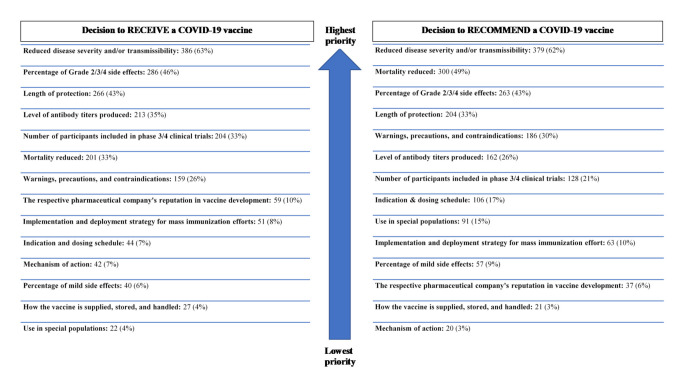
Clinical data prioritized by pharmacists in vaccine decision-making (*n* = 616).

**Table 1 pharmacy-09-00131-t001:** Baseline characteristics of survey respondents (*n* = 632).

Characteristic	All Respondents, *n* (%)
**Regions of the United States**	
Northeast	289 (45.7)
Southeast	124 (19.6)
Southwest	39 (6.2)
Midwest	114 (18)
West	66 (10.4)
**Gender**	
Male	227 (35.9)
Female	396 (62.7)
Transgender man/female-to-male	0 (0)
Transgender woman/male-to-female	0 (0)
Non-binary, genderqueer, or gender nonconforming	1 (0.16)
Other	0 (0)
Prefer not to answer	8 (1.3)
**Age**	
Range, years	23–73
Mean ± Standard Deviation, years	39 ± 12.3
18–55 years	538 (85.1)
>55 years	88 (13.9)
Missing	6 (0.9)
**Race**	
White	427 (67.6)
Asian	115 (18.2)
Black/Latinx/Other	90 (14.2)
**High risk for progression to severe COVID-19 ^1^**	
Yes	136 (21.5)
No	496 (78.5)
**Years as a practicing pharmacist**	
Range	0–50
Mean ± Standard Deviation	13.9 ± 12.3
0–5	203 (32.1)
6–10	126 (19.9)
11–20	139 (22)
21–30	75 (11.9)
31–40	55 (8.7)
41–50	28 (4.4)
Missing	6 (0.9)
**Primary pharmacy practice setting**	
Direct patient care ^2^	434 (68.7)
Indirect patient care ^3^	195 (30.9)
Missing	3 (0.5)
**Specialized therapeutic area ^4^**	
Total pharmacists specialized	407 (64.4)
Ambulatory care	51 (8.1)
Infectious disease/HIV	93 (14.7)
Transplant	1 (0.2)
Oncology	42 (6.6)
Neurology	12 (1.9)
Anticoagulation/cardiology	11 (1.7)
Critical care	29 (4.6)
Emergency medicine	18 (2.8)
Pediatrics	11 (1.7)
Geriatrics	12 (1.9)
Internal medicine	55 (8.7)
Other	72 (11.4)
Total pharmacists not specialized	225 (35.6)

^1^ Cancer, immunocompromised state from bone marrow transplant, immune deficiencies, HIV, use of corticosteroids or other immune weakening medicines, immunocompromised state from solid organ transplant, sickle cell disease, thalassemia, chronic obstructive pulmonary disease, asthma, fibrosis, cystic fibrosis, type 1 diabetes, type 2 diabetes, serious heart condition, heart failure, coronary artery disease, cardiomyopathies, smoker, obesity, cerebrovascular disease, neurologic conditions such as dementia, chronic kidney disease, liver disease. ^2^ Hospital pharmacy, outpatient clinic, community pharmacy. ^3^ Pharmaceutical industry, managed care, academia, pharmacy administration. ^4^ Residency, fellowship, both residency and fellowship.

**Table 2 pharmacy-09-00131-t002:** Pharmacists’ decision for COVID-19 vaccination.

	Vaccinate ≤ 1 Year after Vaccine Approval, *n* (%)	Vaccinate >1 Year after Vaccine after Approval, *n* (%)	*p*-Value *
**5 category comparisons:**			
**Pharmacists’ willingness to RECEIVE a COVID-19 vaccine, *n* = 632**			
Extremely unlikely	83 (13.1)	45 (7.1)	0.00009
Somewhat unlikely	70 (11.1)	50 (7.9)	
Neither likely nor unlikely	55 (8.7)	44 (7)	
Somewhat likely	173 (27.4)	175 (27.7)	
Extremely likely	251 (39.7)	318 (50.3)	
**Pharmacists’ willingness to RECOMMEND a COVID-19 vaccine, *n* = 632**			
Extremely unlikely	50 (7.9)	28 (4.4)	*p* < 0.00001
Somewhat unlikely	64 (10.1)	23 (3.6)	
Neither likely nor unlikely	117 (18.5)	68 (10.8)	
Somewhat likely	206 (32.6)	180 (28.5)	
Extremely likely	195 (30.9)	333 (52.7)	
**2 category comparisons:**			
**Pharmacists’ willingness to RECEIVE a COVID-19 vaccine, *n* = 632**			
Somewhat likely/extremely likely	424 (67.1)	493 (78)	0.000014
Other (neither likely nor unlikely, somewhat unlikely, extremely unlikely)	208 (32.9)	139 (22)	
**Pharmacists’ willingness to RECOMMEND a COVID-19 vaccine, *n* = 632**			
Somewhat likely/extremely likely	401 (63.4)	513 (81.2)	*p* < 0.00001
Other (neither likely nor unlikely, somewhat unlikely, extremely unlikely)	231 (36.6)	119 (18.8)	

* The *p*-value was determined by a chi-squared analysis.

**Table 3 pharmacy-09-00131-t003:** Pharmacists’ perceptions of COVID-19 and COVID-19 vaccines.

	Strongly Disagree	Disagree	Neither Agree nor Disagree	Agree	Strongly Agree
**Perceptions of COVID-19**					
I am concerned about becoming infected with SARs-CoV-2, *n* = 618	27 (4.4)	61 (9.9)	66 (10.7)	305 (49.4)	159 (25.7)
My daily work will be affected if I get COVID-19, *n* = 618	5 (0.8)	28 (4.5)	53 (8.6)	196 (31.7)	339 (54.9)
I have been following the news and literature of COVID-19 vaccines, *n* = 632	9 (1.4)	12 (1.9)	14 (2.2)	293 (46.4)	304 (48.1)
I believe COVID-19 will be a seasonal virus, *n* = 632	116 (18.4)	214 (33.9)	216 (34.2)	69 (10.9)	17 (2.7)
COVID-19 can occur in any season and is dependent on human control to slow the spread, *n* = 632	3 (0.5)	20 (3.2)	38 (6)	243 (38.4)	328 (51.9)
**Perceptions of COVID-19 vaccines**					
I would feel protected if I received a COVID-19 vaccination, *n* = 616	18 (2.9)	56 (9.1)	182 (29.5)	296 (48.1)	64 (10)
I would fear for serious side effects from the COVID-19 vaccine, *n* = 616	51 (8.3)	150 (24.4)	157 (25.5)	183 (29.7)	75 (12.2)
I would doubt or be suspicious about the efficacy of a new COVID-19 vaccine, *n* = 616	42 (6.8)	160 (26)	160 (26)	184 (29.9)	70 (11.4)
I would worry that the FDA overestimated the safety of the COVID-19 vaccine, *n* = 616	39 (6.3)	145 (23.5)	127 (20.6)	214 (34.7)	91 (14.8)

**Table 4 pharmacy-09-00131-t004:** Factors associated with pharmacists’ willingness to receive COVID-19 vaccines.

Variables	Likely to RECEIVE Vaccine ≤ 1 Year from Vaccine Approval, *n* = 632			Likely to RECEIVE Vaccine > 1 Year from Vaccine Approval, *n* = 632			
	Extremely Likely or Somewhat Likely * (*n* = 424)	Extremely Unlikely, Somewhat Unlikely, Neither Likely nor Unlikely * (*n* = 208)	*p*-Value	Extremely Likely or Somewhat Likely * (*n* = 493)	Extremely Unlikely, Somewhat Unlikely, Neither Likely nor Unlikely * (*n* = 139)	*p*-Value	*p*-Value ***
***COVID-19 exposure or personal experience***							
Confirmed diagnosis of COVID-19 **							
Yes	9 (2.1)	5 (2.4)	0.82	10 (2)	4 (2.9)	0.55	0.92
No	415 (97.9)	203 (97.6)		483 (98)	135 (97.1)		
Exposed to a person with COVID-19 based on the CDC definition of exposure							
Yes	90 (21.2)	62 (29.8)	0.46	117 (23.7)	35 (25.2)	0.59	0.66
No	253 (59.7)	106 (51)		285 (57.8)	74 (53.2)		
Unsure	81 (12.8)	40 (6.3)		91 (14.4)	30 (4.7)		
Had a love one fall critically ill or pass away from COVID-19 **							
Yes	50 (11.8)	38 (18.3)	0.27	71 (14.4)	17 (12.2)	0.51	0.24
No	374 (88.2)	170 (81.7)		422 (85.6)	122 (87.8)		
***Primary pharmacy practice setting***							
Direct patient care	275 (64.7)	159 (76.4)	0.004	329 (66.7)	105 (75.5)	0.17	0.55
Indirect patient care	148 (23.4)	47 (7.4)		161 (25.5)	34 (5.4)		
Missing	1 (0.2)	2 (0.3)		3 (100)	0 (0)		
***Background in training***							
Post-doctoral training	290 (68.4)	112 (53.8)		329 (66.7)	73 (52.5)		
No post-doctoral training	134 (31.6)	96 (46.2)	0.00035	164 (33.3)	66 (47.5)	0.002	0.59
***Region of the USA***							
Northeast	178 (42)	106 (51)		217 (44)	67 (48.2)		
Southeast	89 (21)	42 (20.2)		100 (20.3)	31 (22.3)		
Southwest	25 (5.9)	12 (5.8)		29 (5.9)	8 (5.8)		
Midwest	80 (18.9)	32 (15.4)	0.18	90 (18.3)	22 (15.8)	0.67	0.98
West	52 (12.3)	16 (7.7)		57 (11.6)	11 (7.9)		

* All data presented as *n* (%) unless otherwise specified. ** The option of “unsure,” was not an available answer choice for the following question. *** *p*-value compares the somewhat likely, extremely likely at ≤ 1 year vs. > 1 year from vaccine approval.

**Table 5 pharmacy-09-00131-t005:** Factors associated with pharmacists’ willingness to recommend COVID-19 vaccines.

Variables	Likely to RECOMMEND TO PATIENTS ≤ 1 Year from Vaccine Approval, *n* = 632			Likely to RECOMMEND TO PATIENTS > 1 Year after Vaccine Approval, *n* = 632			
	Extremely Likely or Somewhat Likely * (*n* = 401)	Extremely Unlikely, Somewhat Unlikely, neither Likely nor Unlikely * (*n* = 231)	*p*-Value	Extremely Likely or Somewhat Likely * (*n* = 513)	Extremely Unlikely, Somewhat Unlikely, neither Likely nor Unlikely * (*n* = 119)	*p*-Value	*p*-Value ***
***COVID-19 exposure or personal experience***							
Confirmed diagnosis of COVID-19 **			0.82			0.26	0.08


Yes	9 (2.2)	5 (2.2)		13 (2.5)	1 (0.84)		
No	392 (97.8)	226 (97.8)		500 (97.5)	118 (99.2)		
Exposed to a person with COVID-19 based on the CDC definition of exposure			0.11			0.67	0.66




Yes	86 (21.4)	66 (28.6)		123 (24)	29 (24.4)		
No	238 (59.4)	121 (52.4)		295 (57.5)	64 (53.8)		
Unsure	77 (19.2)	44 (19)		95 (18.5)	26 (21.8)		
Had a love one fall critically ill or pass away from COVID-19 **			0.36			0.67	0.76



Yes	52 (13)	36 (15.6)		70 (13.6)	18 (15.1)		
No	349 (87)	195 (84.4)		443 (86.4)	101 (84.9)		
***Primary pharmacy practice setting***			0.02			0.19	0.57

Direct patient care	262 (65.3)	172 (74.5)		346 (67.4)	88 (73.9)		
Indirect patient care	138 (34.4)	57 (24.7)		164 (32)	31 (26.1)		
Missing	1 (0.2)	2 (0.9)		3 (0.6)	0 (0)		
***Background in training***			0.000082			0.0038	0.33
Post-doctoral training	278 (69.3)	124 (53.7)		340 (66.3)	62 (52.1)		
No post-doctoral training	123 (30.7)	107 (46.3)		173 (33.7)	57 (47.9)		
***Region of the USA***			0.03			0.15	0.90
Northeast	167 (41.6)	117 (50.6)		230 (44.8)	54 (45.4)		
Southeast	82 (20.4)	49 (21.2)		100 (19.5)	31 (26.1)		
Southwest	26 (6.5)	11 (4.8)		31 (6)	6 (5)		
Midwest	72 (18)	40 (17.3)		90 (17.5)	22 (18.5)		
West	54 (13.5)	14 (6.1)		62 (12.1)	6 (5)		

* All data presented as *n* (%) unless otherwise specified. ** The option of “unsure,” was not an available answer choice for the following question. *** *p*-value compares the somewhat likely, extremely likely at ≤ 1 year vs. > 1 year from vaccine approval.

## Data Availability

The data presented in this article is not publicly available.

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
