# Peer review of "Pharmacists’ Perceptions and Drivers of Immunization Practices for COVID-19 Vaccines: Results of a Nationwide Survey Prior to COVID-19 Vaccine Emergency Use Authorization"

_pharmacy, 2021, doi:10.3390/pharmacy9030131_

Round 1
Reviewer 1 Report
Reviewed article is very important. Authors presented that among 632 pharmacists approximately 67 to 78% of pharmacists were willing to receive a COVID-19 vaccine and 63 to 81% of them were willing to recommend a COVID-19 vaccine. I understand that more pharmacists recommended vaccine after 1 year from vaccine approval. It is known that the more people get vaccinated and the more we learn about side effects, the more we know about their effectiveness. However, there are about 20-30% of pharmacists who will not vaccinate themselves and will not recommend vaccination. In my country, I see a similar relationship that doctors recommend vaccinations against e.g. measles, hepatitis B, but the same doctors are sometimes against SARS-CoV-2 vaccinations. This is an important problem that should be described in more detail in the Discussion, or even as a separate sub-item of the Discussion. I did not find it in the text, but the authors should specifically write which groups of pharmacists are "against" vaccination, whether it is related to, for example, age, race or years of practice. Moreover, in Introduction, Authors should add some general information about vaccines against SARS-CoV-2, it is about types of vaccines and which are already in used and how many were in 2020, and are today in clinical phases. Please cite the following articles: https://www.thno.org/v10p7821 and https://www.thno.org/v11p1690.htm
Author Response
Thank you for your review and feedback. I have updated the manuscript to emphasize what groups of pharmacists were less likely than likely to receive/recommend COVID-19 vaccines in relation to personal experience/exposure to COVID-19, pharmacy practice setting, and background in training since these were our primary outcomes. Willingness to receive/recommend in relation to differences in age, gender, race, and years of practice was not studied. I also included general information in the introduction about COVID-19 vaccine landscape during the beginning of 2020 and the vaccine landscape now, focusing on the 2 mRNA vaccines and viral vector vaccines in the introduction because those were the only ones approved for emergency use authorization in the United States by the FDA. I chose to use the World Health Organization and an article from Nature (instead of https://www.thno.org/v10p7821 and https://www.thno.org/v11p1690.htm ) because these references were updated (especially the WHO) link were updated recently and continiously. Thank you.

Reviewer 2 Report
Congratulations for the interesting article.It was a pleasure reading it. I would suggest the authors to include articles in the discussion chapter and compare their results with other published papers.
In my opinion this is a well written article. The authors were interested in the role of the Pharmacists in recommending and providing vaccines to improve public health. Because the authors performed a cross-sectional survey with a big number of respondents, 632 participants , the data can be relevant statistically speaking. The data were analyzed by using descriptive and inferential statistics so the results were relevant. I would suggest the authors to include more recent articles in the discussion chapter and compare their results with other published papers because in the scientific literature the authors can find multiple articles on this topic regarding the vaccination process. My idea of including more recent articles was because now a lot of articles are published that regard the vaccination and because this is still a hot topic.
Author Response
Thank you for your thoughtful review. After conducting a literature search, I found 2 additional relevant article by Merks et al and Daly et al that I included in the discussion of the manuscript. However, to my knowledge no other studies have been published to date about pharmacists' perceptions and willingness to receive and recommend COVID-19 vaccines. There have been studies about pharmacists' willingness to provide other COVID-19 services like testing but not vaccines.
